# High Prevalence and Persistence of *Escherichia coli* Strains Producing Shiga Toxin Subtype 2k in Goat Herds

Xi Yang,[a] Qian Liu,[a] Xiangning Bai,[a,b,c] Bin Hu,[d] Deshui Jiang,[e] Hongbo Jiao,[e] Liangmei Lu,[e] Ruyue Fan,[d] Peibin Hou,[d] Andreas Matussek,[b,c,f] Yanwen Xiong[a]

[a]State Key Laboratory of Infectious Disease Prevention and Control, National Institute for Communicable Disease Control and Prevention, Chinese Center for Disease Control and Prevention, Beijing, China
[b]Division of Clinical Microbiology, Department of Laboratory Medicine, Karolinska Institute, Stockholm, Sweden
[c]Division of Laboratory Medicine, Oslo University Hospital, Oslo, Norway
[d]Shandong Center for Disease Control and Prevention, Jinan, Shandong, China
[e]Lanling Center for Disease Control and Prevention, Lanling, Shandong, China
[f]Division of Laboratory Medicine, Institute of Clinical Medicine, University of Oslo, Oslo, Norway

**ABSTRACT**  Shiga toxin (Stx)-producing *Escherichia coli* (STEC) is a zoonotic pathogen with the ability to cause severe diseases like hemorrhagic colitis (HC) and hemolytic uremic syndrome (HUS). Shiga toxin (Stx) is the key virulence factor in STEC and can be classified into two types, Stx1 and Stx2, and different subtypes. Stx2k is a newly reported Stx2 subtype in *E. coli* strains from diarrheal patients, animals, and raw meats exclusively in China so far. To understand the reservoir of Stx2k-producing *E. coli* (Stx2k-STEC), we investigated Stx2k-STEC strains in goat herds and examined their genetic characteristics using whole-genome sequencing. A total of 448 STEC strains were recovered from 2,896 goat fecal samples, and 37.95% (170/448) were Stx2k-STEC. Stx2k-STEC strains of serotype O93:H28 and sequence type 4038 (ST4038) were the most predominant and were detected over several years. Notably, 55% of Stx2k-STEC strains carried the heat-labile toxin (LT)-encoding gene (*elt*) defining enterotoxigenic *E. coli* (ETEC), thereby exhibiting the hybrid STEC/ETEC pathotype. Stx2k-converting prophage genomes clustered into four groups and exhibited high similarity within each group. Strains from patients, raw meat, sheep, and goats were intermixed distributed in the phylogenetic tree, indicating the risk for cross-species spread of Stx2k-STEC and pathogenic potential for humans. Further studies are required to investigate the Stx2k-STEC strains in other reservoirs and to understand the mechanism of persistence in these hosts.

**IMPORTANCE**  Strains of the recently reported Stx2k-STEC have been circulating in a variety of sources over time in China. Here, we show a high prevalence of Stx2k-STEC in goat herds. More than half of the strains were of the hybrid STEC/ETEC pathotype. Stx2k-STEC strains of predominant serotypes have been widespread in the goat herds over several years. Stx2k-converting prophages have exhibited a high level of similarity across geographical regions and time and might be maintained and transmitted horizontally. Given that goat-derived Stx2k-STEC strains share similar genetic backbones with patient-derived strains, the high prevalence of Stx2k-STEC in goats suggests that there is a risk of cross-species spread and that these strains may pose pathogenetic potential to humans. Our study thus highlights the need to monitor human Stx2k-STEC infections in this region and, by extension, in other geographic locations.

**KEYWORDS**  *Escherichia coli*, Shiga toxin (Stx), Stx2k subtype, Stx2k-converting prophage, whole-genome sequencing

Address correspondence to Yanwen Xiong, xiongyanwen@icdc.cn.

The authors declare no conflict of interest.

Shiga toxin (Stx)-producing *Escherichia coli* (STEC) is a significant zoonotic foodborne pathogen worldwide. The clinical manifestations of STEC infection range from asymptomatic carriage, diarrhea, or hemorrhagic colitis (HC) to the potentially life-threatening complication hemolytic uremic syndrome (HUS), which can lead to acute renal failure (1). Ruminants are important reservoirs of STEC, and although cattle have been recognized as the primary source of STEC, recent epidemiological studies have suggested that goats and sheep are also significant sources of human STEC infection (2–4). Human STEC infection occurs through direct or indirect contact with animals, e.g., animal slaughter processes or exposure to fecal-matter-contaminated water, soils, animal-derived food, and agricultural products.

STEC pathogenicity is mainly due to the production of one or more kinds of Shiga toxins (Stxs), which are encoded in the late region of lysogenized lambdoid prophages. Stx-converting prophages, as horizontal gene transfer (HGT) elements, can convert a harmless commensal into an enteric pathogen and promote the emergence of hybrid pathotypes (5), thereby playing an important role in STEC pathogenesis and evolution. Stx is an $AB_5$ toxin and includes two antigenic forms, Stx1 and Stx2. Using the nomenclature proposed by Scheutz et al., several subtypes of Stx1 (Stx1a, Stx1c, and Stx1d) and Stx2 (Stx2a to Stx2g) have been described (6). Based on epidemiological studies, different Stx subtypes vary in toxicity, leading to different patient outcomes (7). Stx2-producing strains are associated with more severe disease than Stx1-producing strains, particularly strains harboring Stx2a and Stx2d subtypes (8). In recent years, new Stx2 subtypes have been identified: Stx2h to Stx2m and Stx2o (8–13). Of these, Stx2j, Stx2k, Stx2m, and Stx2o subtypes have been identified in clinical isolates (11, 13), suggesting clinical relevance of new Stx subtypes. Interestingly, Stx2k-STEC strains have been identified in *E. coli* strains from a variety of hosts and sources, including diarrheal patients, animals (goats, sheep, and pigs), and raw meat (mutton and beef), in China over time (12, 14), while they not yet been reported in any other countries.

In this study, we investigated the Stx2k-STEC strains in goat herds as one important reservoir through our continuous STEC monitoring over years. Whole-genome sequencing (WGS) was used to characterize Stx2k-STEC strains, including their molecular traits and Stx2k-converting phages and the genetic relatedness of strains from diverse sources.

## RESULTS

**High prevalence of Stx2k-STEC strains in goat herds.** In total, 448 STEC strains were isolated from 2,896 goat fecal samples, giving a culture-positive rate of 15.47% (448/2,896). The rates of isolation of STEC strains from goat feces during the study period are shown in Table 1. A total of 12 different *stx* subtypes/combinations were identified among 448 STEC isolates, with *stx2k* (169/448, 37.72%) being the most predominant, followed by *stx1c* (120/448, 26.69%) and *stx1c+stx2b* (65/448, 14.51%). One isolate carried *stx2k* and *stx2e*. The proportions of *stx2k*-positive strains were highest in 2019 and 2021 (Fig. 1).

Among the 170 *stx2k*-positive STEC strains, 167 shared an identical *stx2k* sequence with the previously reported Stx2k-STEC strain STEC309 (GenBank accession number CP041435.1) (12), while the *stx2k* sequence harbored by the remaining three strains differed from that of strain STEC309 at position 346 in the A subunit, leading to a change in one amino acid (from D to N). The B subunit and the intergenic region between the A and B subunits (AGGAGTTAAGT) were highly conserved among all goat-derived Stx2k-TEC strains in this study.

**Molecular characteristics of goat-derived Stx2k-STEC isolates.** Among the 170 goat-derived Stx2k-STEC strains, 16 different O:H serotypes were found. O93:H28 was the most predominant serotype, accounting for 30.59% (52/170) of all strains (Table S1 in the supplemental material). Two novel O serogroup genotypes (OgN-RKI3:H21 and OgN17:H21) were found among 34 Stx2k-STEC isolates (20.0%) (Table 2). Sixteen multilocus sequence types (MLST) were assigned among all isolates; of these, sequence type 4038 (ST4038) was the most predominant (52/170, 30.59%). Isolates with the same serotype were assigned as the same sequence type, with the exceptions of OgN-RKI3:H21 (ST683 and ST58) and O116:H25 (ST155 and ST58) (Table 2). The dominant

**TABLE 1** Prevalence of STEC in goat herds from 2017 to 2021

| Yr | No. (%) of: | | |
|---|---|---|---|
| | Samples | STEC isolates[a] | Stx2k-STEC strains[b] |
| 2017 | 407 | 83 (20.39) | 20 (24.10) |
| 2018 | 325 | 32 (9.85) | 12 (37.5) |
| 2019 | 512 | 86 (16.80) | 37 (43.02) |
| 2020 | 646 | 108 (16.72) | 36 (33.33) |
| 2021 | 1,006 | 139 (13.82) | 65 (46.76) |
| Total | 2,896 | 448 (15.47) | 170 (37.95) |

[a]Culture-positive rate of STEC strains among all samples.
[b]Prevalence of Stx2k-STEC among all STEC isolates.

serotype (sequence type) in 2017 and 2019 was O93:H28 (ST4038), whereas in 2018 and 2021, OgN-RKI3:H2 (ST683) predominated.

A total of 93 virulence genes were identified among 170 Stx2k-STEC strains based on the Virulence Factor Database (VFDB). These virulence genes can be classified into

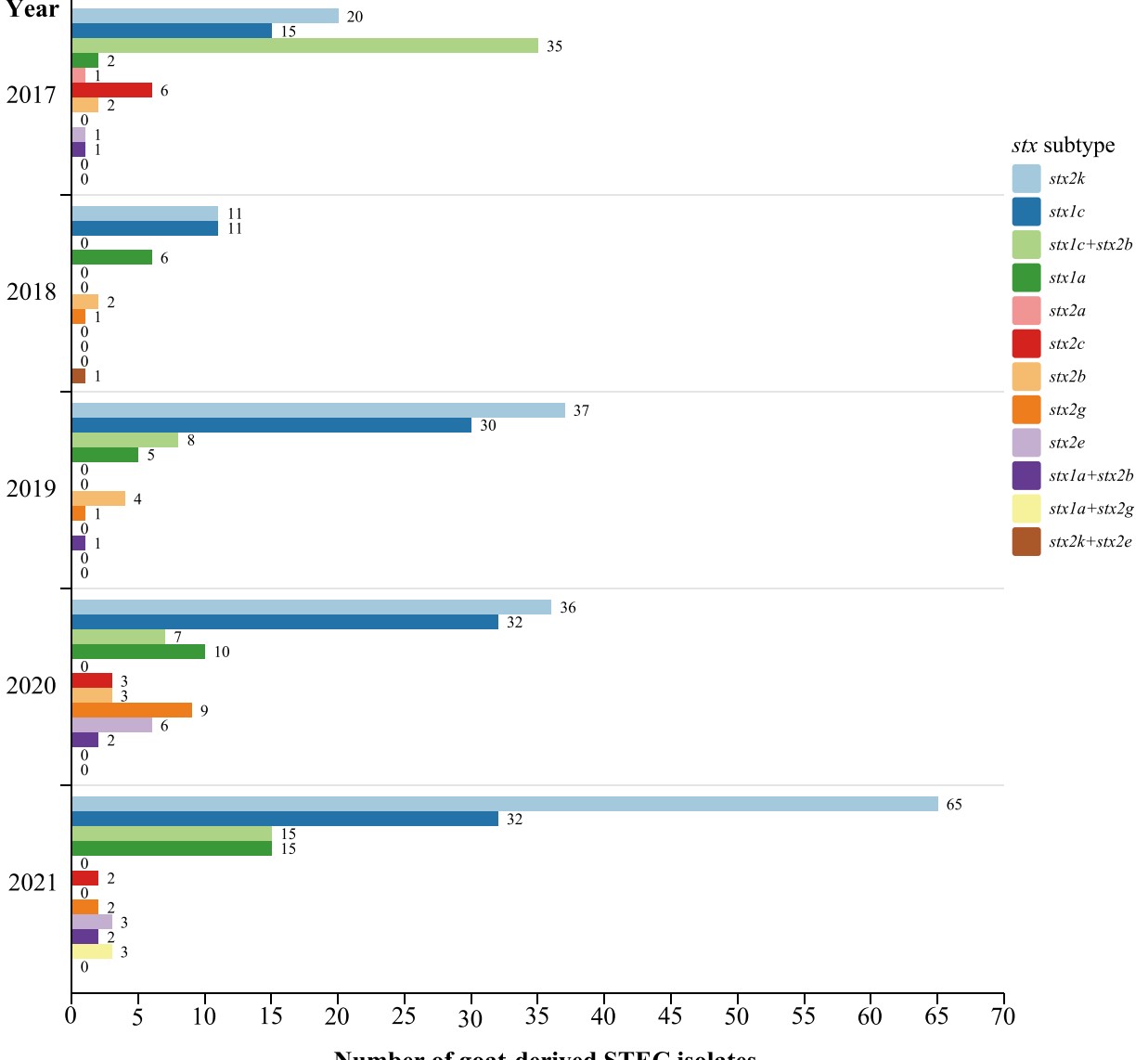

**FIG 1** Distribution of Shiga toxin subtypes from 2017 to 2021.

**TABLE 2** Serotypes and sequence types of the 170 Stx2k-STEC isolates from 2017 to 2021

| Serotype | MLST(s) (no. of isolates) | No. of isolates obtained in: | | | | |
|---|---|---|---|---|---|---|
| | | 2017 | 2018 | 2019 | 2020 | 2021 |
| O93:H28 | ST4038 (52) | 12 | | 31 | 8 | 1 |
| OgNRKI3:H21 | ST683 (30), ST58 (1) | | 10 | 1 | 2 | 18 |
| O184:H19 | ST6313 (17) | | | | | 17 |
| O174:H2 | ST13029 (13) | | | | | 13 |
| O22:H8 | ST446 (12) | 3 | | 1 | 3 | 5 |
| O133:H25 | ST10326 (11) | | 1 | 2 | 8 | |
| O16:H32 | ST1286 (8) | | | | 8 | |
| O22:H16 | ST295 (5) | | | | | 5 |
| O116:H25 | ST155 (3), ST58 (1) | 2 | | 1 | 1 | |
| O8:H19 | ST162 (4) | | | | 4 | |
| OgN17:H21 | ST602 (3) | | | 1 | 2 | |
| O100:H19 | ST1611 (3) | | | | | 3 |
| O159:H25 | ST155 (3) | | | | | 3 |
| O8:H53 | ST345 (2) | 2 | | | | |
| O86:H51 | ST155 (1) | 1 | | | | |
| O48:H21 | ST5221 (1) | | 1 | | | |
| Total | 170 | 20 | 12 | 37 | 36 | 65 |

several groups based on their functions. Ninety-four Stx2k-STEC strains (55.3%) in this study carried the heat-labile toxin (LT)-encoding gene, a virulence determinant for entero-toxigenic *E. coli* (ETEC), thereby exhibiting the hybrid STEC/ETEC pathotype. All of the STEC/ETEC strains carried the heat-labile toxin-encoding gene *elt-II* (Table S1). Other virulence genes were involved in adherence (*fdeC*, *cfaABCDE*, *cgsABCDEFG*, *fimABCDEFGHL*, and *ecpABCDER*), invasion (*aslA*, *ibeB*, *ompA*, and *AAA92657*), iron uptake (*ybtS*, *fepABCDEG*, and *fur*), secretion system effectors (*yopP*, *espLRXY*, *fha rhs*, *tssABCDFGHJKLM*, and *gspCDE-KLM*), regulation (*phoP*, *rcsB*, and *rpoS*), etc. (Table S2a). Other important virulence factors of STEC, e.g., intimin encoded at the locus of enterocyte effacement (LEE) pathogenicity island, were absent. Furthermore, the major colonization factor (CF)-encoding genes of ETEC, such as *faeG* (F4, previously termed K88), *fedA* (F18), *fanC* (F5, K99), and *fasA* (F6, 987P), were not detected.

Antimicrobial resistance (AMR) genes associated with resistance to peptide antibiotics (*pmrF*) and class C $\beta$-lactamases (*bla*$_{EC}$, *ampH*, and *ampC1*) were found in all strains. Other predominant AMR genes were *tet*(B) ($n = 29$ strains), *aph(6)-Id* ($n = 17$), *qnrS1* ($n = 16$), *bla*$_{TEM-181}$ ($n = 16$), and *sul3* ($n = 16$) (Table S2b).

**Genetic features of Stx2k-converting prophages.** Twelve intact and 158 incomplete Stx2k-converting prophages were identified from 170 Stx2k-STEC genomes. We assessed the genetic relationships among all of the Stx2k-converting prophages in this study and reference Stx2k-STEC strains reported previously. Six clusters of prophages were obtained by using VIRIDIC (Virus Intergenomic Distance Calculator) (Fig. S1a). Clusters 1 and 4, sharing the same downstream or upstream gene and similar structures, were defined as phage groups, as were clusters 2 and 6, and thus, four Stx2k pro-phage groups, termed Stx2k prophages G1, G2, G3, and G4, were designated (Fig. S1b). Stx2k prophages in this study ($n = 170$) were distributed in G1 to G3, and one Stx2k prophage from a reference pig-derived STEC strain was grouped as G4 (Fig. S1). Prophages identified from draft genomes with lengths of ≤8,000 bp were assigned as "unknown group" (Gx). Three different integrases were identified in 44 prophages (data not shown), and consistently, three insertion sites were found, i.e., *yccA* (modula-tor of FtsH protease YccA), *dusA* (tRNA-dihydrouridine synthase A), and *yecE* (DUF72 domain-containing protein YecE) (Table S1). The integrases and insertion sites corre-sponded to the three Stx2k prophage groups (Fig. 2).

To characterize and compare the Stx2k-converting prophages from strains of differ-ent molecular characteristics, Stx2k-STEC strains representative of different serotypes

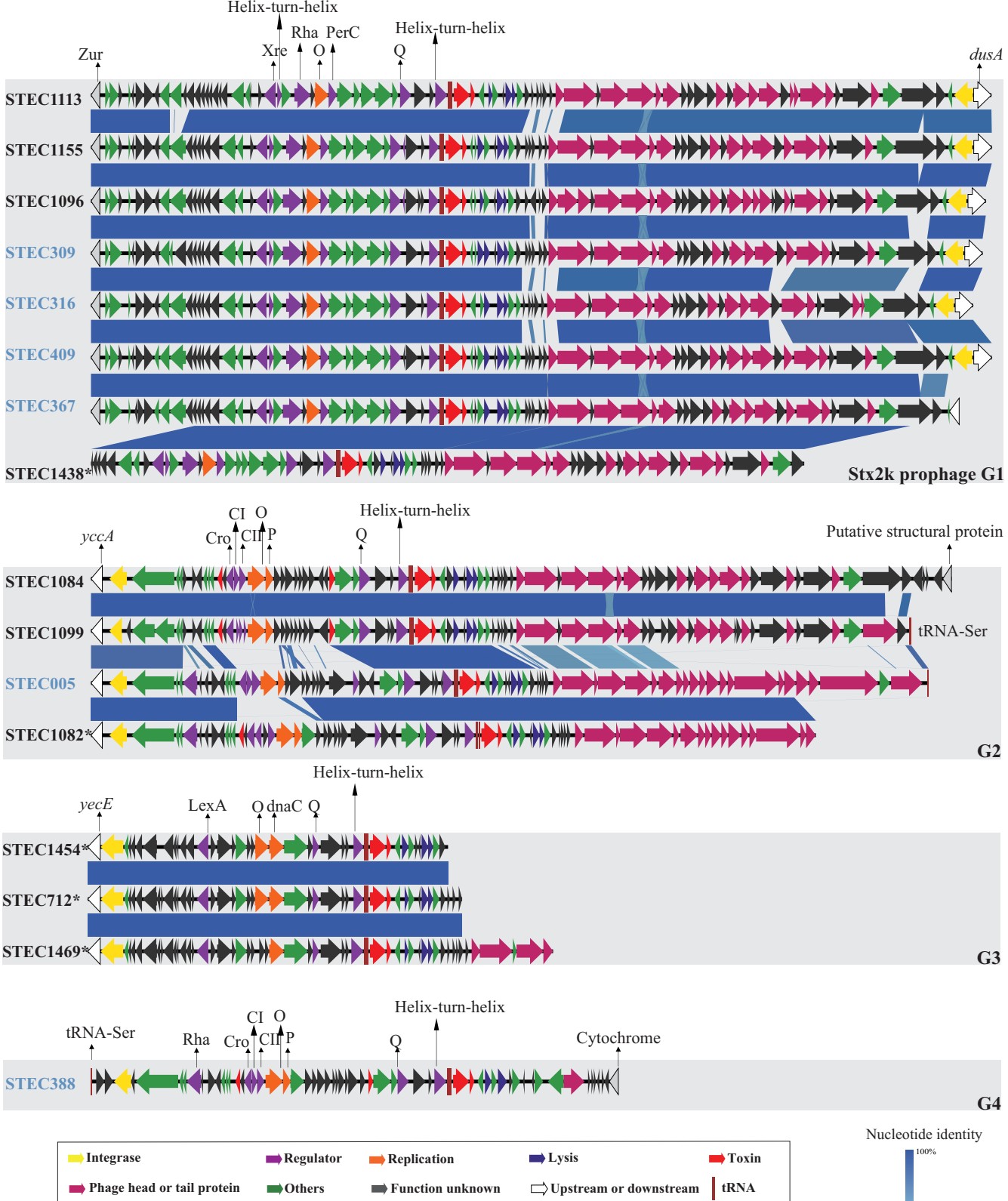

**FIG 2** Easyfig plot comparing Stx2k-converting prophages from Stx2k-STEC strains representative of different serotypes and sequence types within each prophage group. Arrows indicate gene directions. Coding sequences are represented by arrows and linked by blue bars shaded to represent the nucleotide identity, as indicated in the key. The colors of the strain designations indicate the sources of strains: black represents goat-derived strains in this study, and light blue represents strains derived from other sources in other studies. An asterisk (*) signifies an incomplete prophage. Prophages belonging to the same groups are shown in gray-shaded boxes, and the Stx2k prophage group is shown in the bottom-right corner of each box.

and sequence types were selected for further comparisons. A progressiveMauve alignment revealed the blocks with collinearity between Stx2k-converting prophages (Fig. S2). Stx2k prophages of the same group displayed nearly identical collinear-block components. Consistent results were observed using Easyfig. The Stx2k-converting prophages were generally organized into three major modules, i.e., an integration cassette, lysis cassette functions, and morphogenesis-related genes. The Stx2k-converting prophages aligned across their lengths with few structural variations in each group. For instance, the insertion sites, integrases, replication proteins, and antiterminator protein Q of the phages are the elements that are the same in each prophage group (Fig. 2). All of the Stx2k prophages shared a conserved region of approximately 40 kb surrounding *stx*. The prophage structures among those groups mainly differed in their early regions (upstream from *stx2k*), especially the genes encoding regulator and replication functions, while only minor structural variations were observed in their late regions, encoding the morphogenetic and lysis cassette functions.

**Phylogenetic analysis.** To determine the phylogenetic relationships of Stx2k-STEC strains isolated from goats in this study and 17 previously reported Stx2k-STEC strains from diverse sources (12, 14), a whole-genome phylogeny tree was constructed from the alignment of the concatenated coding sequences (CDSs) of the 1,909 shared loci found in all 187 Stx2k-STEC isolates (Fig. 3). Strains isolated from different years were intermixed in the phylogenetic tree. Strains belonging to the same serotype and MLST type showed the tendency to cluster together. Six phylogenetic clusters comprising strains isolated from different sources or geographical locations were observed. Pairwise single-nucleotide polymorphism (SNP) distance heatmaps were produced to illustrate the dissimilarity of strains within each cluster (Fig. S3). Two large clusters were observed, comprising 36% and 31% of isolates, respectively, and were referred to as cluster 1 and cluster 2. Cluster 1 primarily consisted of 47 OgN-RKI3:H21 isolates from goats, sheep, and patients from three different provinces, with the patient-derived strain and the other isolates differing in 292 to 720 SNPs (Fig. S3a). All isolates from cluster 2 belonged to serotype O93:H28 and were isolated from goats in this study. All isolates in cluster 3 belonged to serotype O100:H19 and were isolated from goats and raw meat, with strains from the two sources differing in 62 to 265 SNPs (Fig. S3b). Goat-derived STEC strains isolated from two provinces were grouped in cluster 4 and differed in 458 to 459 SNPs (Fig. S3c). Strains within cluster 5 belonged to different serotypes and were more diversely distributed, with the differences between strains from the two sources in cluster 5 (patients and goats) ranging from 8,915 to 9,984 SNPs (Fig. S3d). One pig-derived O159:H16 STEC isolate formed a separate cluster. Different Stx2k prophage groups were observed within the same phylogenetic cluster (Fig. 3).

## DISCUSSION

Stx2k-producing *E. coli* strains have been identified from a broad range of hosts and sources, including diarrheal patients, pigs, raw meat (beef and mutton), sheep, and goats, exclusively in China so far (12, 14). In this study, 170 Stx2k-STEC strains were isolated from goats, and the proportion of Stx2k (37.95%) was the highest among all Stx subtypes detected. Goats have been identified as significant reservoirs of STEC and are capable of carrying STEC without clinical signs themselves (3, 4). Previous studies have suggested that the most frequent *stx* type encountered in goat-derived isolates was *stx1* (15–17), especially subtype *stx1c* (18, 19). This might be due to the lack of systematic investigation of STEC among goats (4). Also, since the application of WGS is becoming routine for STEC genotyping, the diversity of *stx* subtypes in STEC strains from different sources is accumulating. Although Stx2k-STEC has only been reported in China so far, it is not unlikely that Stx2k-STEC may occur in other countries but has not been identified or has been assigned as another Stx2 subtype; for instance, Stx2k was initially assigned as a new Stx2e variant (20). One example is that the recently designated Stx2h subtype was previously only detected in isolates from Tibetan marmots (9) but has been reported recently from sprouts in Canada (11). The host ranges and specificities of different *stx* subtypes need further study.

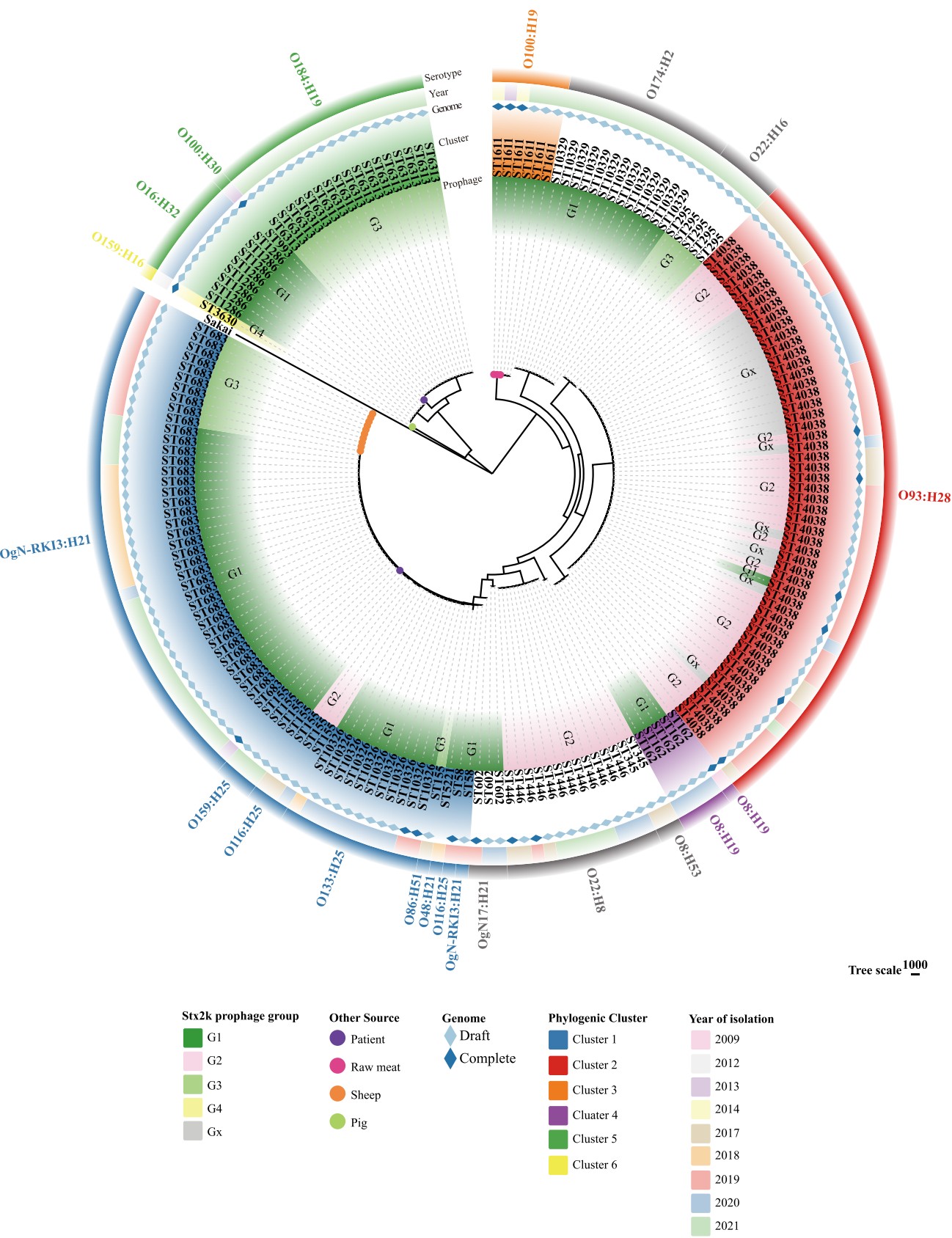

**FIG 3** Whole-genome phylogeny of Shiga toxin 2k-producing *E. coli* (Stx2k-STEC) isolates. Circular representation of the Gubbins phylogenetic tree generated from the concatenated sequences of the shared loci found in the wgMLST analysis. The Gubbins tree was annotated with relevant metadata using the online tool ChiPlot (https://www.chiplot.online/). From the outer to the inner, the circles represent serotype, year of isolation, genome completeness, phylogeny cluster, Stx2k prophage group, and isolation source. The leaves of the tree indicate the MLST sequence type.

Previous studies have reported differences between Stx1/Stx2 types and subtypes in terms of host specificity and binding strength (6, 21). One Stx2k variant was reported to be less toxic than Stx2a but similar in receptor-binding preference, acid tolerance, and thermostability in a Vero cell assay (22). It has been proposed that the two amino acid differences in the receptor binding site might explain the difference in cytotoxicity between Stx2a and Stx2k. The similarity between Stx2k and Stx2a and the high prevalence of Stx2k-STEC in goat herds suggest a possible risk of Stx2k-STEC causing human infections. In addition to three *stx2k* variants previously reported (12), a new *stx2k* variant was identified in this study; thus, there are four variants of Stx2k to date. It remains to be investigated whether the amino acid changes between different variants in the catalytically active A subunit play a role in the toxicity of Stx2k.

The WGS-based molecular characterization revealed that O93:H28 was the most predominant serotype in goat-derived Stx2k-STEC strains. To the best of our knowledge, no human disease has been reported to be associated with this serotype thus far. The reason for this may be that most serotyping is performed by conventional slide agglutination tests using antisera against the recognized *E. coli* O antigens established by the International Centre for Reference and Research on *Escherichia* and *Klebsiella*, and the former scheme did not include O93 (23, 24). The novel O serogroup genotype OgN-RKI3 has exclusively been reported in human-derived STEC strain 16-04178 (GCA_016116345.1) carrying subtype *stx2e* (25, 26). Our finding that 18% of goat-derived Stx2k-STEC strains in this study possessed the same O genotype as strain 16-04178 indicated that OgN-RKI3 strains have a potentially wide distribution. Remarkably, 55% of Stx2k-STEC strains in our study carried the heat-labile toxin (LT)-encoding gene *elt*, exhibiting a STEC/ETEC hybrid pathotype. In a recent study, 56% of Stx2e-STEC strains exhibited the STEC/ETEC hybrid pathotype (27). The colonization factors (CFs) and colonization factor antigens (CFAs) of ETEC strains confer host specificity (28). The goat-derived STEC/ETEC hybrid strains in this study lacked animal-specific CFs, such as K88 and K99 (29), whereas some strains carried human-derived ETEC CFA-encoding genes. The difference in major adhesin factors might influence the intestinal colonization of strains, and thus, further investigation is needed. Notably, Stx2k was initially assigned as a new Stx2e variant from its type strain STEC388, isolated from a healthy pig (20). Stx2k- and Stx2e-producing STEC strains may share similar genetic backbones that can drive the transmission of genes among strains, facilitating the emergence of hybrid pathotypes; further research is needed to elucidate this.

The clustering of Stx2k-converting prophages demonstrated the similarity among Stx2k prophages. This was consistent with a previous study indicating that Stx2a phages exhibited overall sequence similarities to each other (30) and that the Stx2c phages of the O157 strains were highly homogeneous (31, 32). In contrast, Stx2e prophages exhibited considerable diversity (27, 33). In this study, Stx2k-converting phages from diverse sources or geographical regions were distributed in three designated prophage groups. Comparison of Stx2k-converting prophages in terms of genomic organization and the nucleotide sequences of homologous genes highlighted the similarity within each group. Stx phage integrases determined the reorganization of different insertion sites within the bacterial chromosome (5). Among the three insertion sites in the goat-derived Stx2k-STEC strains, *yecE* was commonly found in Stx prophages (27, 33–35), whereas the other two insertion sites were rarely reported. *yccA*, associated with the Sp4 prophage in O157:H7 (31), was found as the insertion site in Stx2e-, Stx2k- and Stx2a-converting prophages (12, 27, 36). The insertion site was usually located upstream from the phage, but in the predominate group G1, insertion site *dusA* was located downstream from the phage and was found in Stx2k-, Stx2a- and Stx2d-converting prophages (12, 36). This tRNA-dihydrouridine synthase also carries a phage scar in O157:H7 (31). Our study showed that the genomic diversity between each Stx2k phage group mainly occurred at their early regions, which contained regulator- and replication-related genes. This region mediates the switch between repression and induction of the prophage (37) and, thus, may influence the expression level of the Shiga toxin (30), as well as the potential to produce phage particles without inducer agents of STEC strains (38). It has been reported that Stx2k

could be measured both in a Vero cell cytotoxicity assay and in a commercial lateral flow immunoassay (12). Further studies are needed to detect the toxin levels of STEC strains carrying each Stx2k prophage group. The differences between Stx2k prophage groups and the presence of conserved regions highlighted the modular structure of lambdoid phages (39).

The whole-genome phylogeny of all goat-derived Stx2k-STEC strains in this study demonstrated considerable genetic diversity. The dominant clusters (cluster 1, OgN-RKI3: H21 and ST6830, and cluster 2, O93:H28 and ST4038) were widespread in this region in the goat herds over years. A high degree of genetic similarity was observed between Stx2k-STEC strains from goats and those reported from other sources. One patient-derived strain in cluster 1 (OgN-RKI3:H21 and ST6830) and goat-derived strains clustered phylogenetically closely, suggesting a pathogenic potential of goat-derived Stx2k-STEC strains. Further studies are essential to investigate the prevalence of Stx2k-STEC strains among humans in this region and to assess their pathogenic potentials. The genomic similarity between the Stx2k-STEC isolates, despite the differences in their timing of isolation, geographic locations, and hosts and sources of isolation, may indicate a wide spread of these strains across species and in different regions in China. Further studies are required to investigate Stx2k-STEC strains in diverse sources, especially humans, and in other geographical regions in China and abroad. Furthermore, the distribution of Stx2k prophage groups in the phylogeny tree demonstrates that the Stx2k prophage does not parallel the whole-genome phylogeny, suggesting that the Stx2k phage and its bacterial host evolve independently. Strains in the same clusters might descend from one ancestor and acquire the Stx2k prophages in independent transduction events.

In conclusion, this study reports a high prevalence of the newly reported Stx2k subtype in STEC strains in goat herds. Strains of predominant serotypes and Stx2k-converting prophage groups were recovered over years. The goat-derived Stx2k-STEC strains in this study demonstrated genetic similarity to those from other sources. Thus, possible Stx2k-STEC transmission to and pathogenic potential for humans should be noted.

## MATERIALS AND METHODS

**Ethics statement.** The current study has been reviewed and approved by the ethics committee of the National Institute for Communicable Disease Control and Prevention, China CDC, with the number ICDC-2017006. Fecal samples of goats were acquired with the consent of the owners of the animals.

**Sample collection and isolation of STEC strains.** In total, 2,896 fecal samples from healthy-looking goats were collected from different family-scale farms in Lanling County, Shandong Province, China, from November 2017 to October 2021. Fecal samples were collected in 2-mL sterile tubes containing Luria-Bertani (LB) medium in 30% glycerol and transported in bags of ice to the laboratory at the National Institute for Communicable Disease Control and Prevention, China CDC, for the isolation of STEC. Strains were isolated and confirmed by methods described previously (40). Briefly, after enrichment with *E. coli* broth (EC broth, Land Bridge, Beijing, China), the samples were examined by PCR for the presence of *stx* (41). Samples positive for *stx* were inoculated onto CHROMagar ECC agar (CHROMagar, Paris, France). After overnight incubation at 37°C, green-blue or colorless colonies on agar were picked and tested for *stx* by a single-colony duplex PCR assay (40). API 20E biochemical test strips (bioMérieux, Lyon, France) were used to confirm that all *stx*-containing isolates were *E. coli*.

**WGS and genome assembly.** Bacterial DNA extraction was performed with the Wizard genomic DNA purification kit (Promega, Madison, WI, USA) according to the manufacturer's protocol. Library preparation and whole-genome sequencing (WGS) were performed at Beijing Novogene Bioinformatics Technology Co., Ltd., China. To obtain the draft genomes of STEC, DNA library preparation was done using the NEBNext Ultra DNA library prep kit (New England Biolabs, Ipswich, MA, USA). The library was then paired-end (2 × 150 bp) sequenced using the Illumina NovaSeq platform (Illumina, San Diego, CA, USA). The low-quality reads (quality score of <Q20) were filtered by using fastp 0.20.1 (https://github.com/OpenGene/fastp) and then *de novo* assembled using SKESA version 2.4.0 (https://github.com/ncbi/SKESA), and low-quality contigs (lengths of <500 bp) were filtered with SeqKit version 0.11.0. To obtain the complete genomes of selected Stx2k-STEC strains, two sequencing libraries were prepared: besides the Illumina DNA library, a 10-kb library was done using a SMRTbell template prep kit (version 1.0). The long library was sequenced using the PacBio Sequel platform (Pacific Biosciences, Menlo Park, CA, USA). The long reads were preliminarily processed for quality control using the "RUN QC" module in SMRT Link version 5.1.0 (www.pacb.com/support/software-downloads), *de novo* assembled using the Hierarchical Genome Assembly Process (HGAP) pipeline, and then corrected using the Illumina short reads to get complete genomes. All genomes were annotated with Prokka (version 1.13.3) (42).

**Molecular characterization of STEC isolates.** Molecular characterization was based on WGS data, including *stx* subtyping, serotyping, multilocus sequence typing (MLST), and detection of virulence factor genes

and antimicrobial resistance genes. Briefly, an in-house *stx*_subtyping database, including representative nucleotide sequences of all identified *stx1* (*stx1a*, *stx1c*, and *stx1d*) and *stx2* (*stx2a* to *stx2m* and *stx2o*) subtypes, was created, and assemblies were compared against the *stx*_subtyping database using ABRicate version 0.8.10 (https://github.com/tseemann/abricate) to determine the *stx* subtypes. Similarly, a serotyping database was created, including all known *wzx/wzy* sequences from typical *E. coli* O serogroups from O1 to O188, as well as *fliC* and its homologous sequences from typical *E. coli* H types from H1 to H56, downloaded from the Center for Genomic Epidemiology (CGE) (database version 1.0.0) (https://bitbucket.org/genomicepidemiology/serotypefinder_db/src/master/) (43), and a set of recently defined novel O antigen biosynthesis gene cluster types (Og types) of *E. coli* (25, 44). MLST of seven housekeeping genes was performed through an on-line tool provided by the Warwick *E. coli* MLST scheme website (https://enterobase.warwick.ac.uk/species/ecoli/allele_st_search). Detection of virulence genes and antimicrobial resistance genes was performed by comparing assemblies against the Virulence Factor Database (VFDB) (45) and the Comprehensive Antibiotic Resistance Database (http://arpcard.mcmaster.ca), respectively, using ABRicate with the default parameters (coverage of ≥80% and identity of ≥80%). Given that, according to a previous study, some Stx2k-STEC strains carried heat-stable and heat-labile enterotoxin-encoding genes (*est* and *elt*) (12), we further subtyped *est* and *elt* genes by comparing the assemblies against an in-house database, including representative nucleotide sequences of *est* and *elt*, using ABRicate version 0.8.10. Nucleotide sequences of different *est* and *elt* subtypes were obtained from previous reports (46–48).

**Identification and characterization of Stx2k-converting prophages.** Stx2k prophage sequences were extracted from genomes and characterized using methods described previously (27). Briefly, PHAge Search Tool Enhanced Release (PHASTER, http://phaster.ca/) was used to identify Stx prophages, and intact and incomplete Stx2k prophage sequences were extracted from complete and draft genomes, respectively. The RAST server (http://rast.nmpdr.org/) was used to annotate the genomes of Stx prophages. The phage insertion site was defined as the gene adjacent to the integrase (33). To assess the genetic relationships of Stx2k prophages, hierarchical clustering of prophages was calculated based on genome alignments and intergenomic similarities between pairs of prophages using VIRIDIC (Virus Intergenomic Distance Calculator) with the default parameters (49). Stx2k-converting prophage clusters sharing an identical downstream or upstream gene and similar structures were recognized as the same group. The sequences of Stx2k prophages were compared and visualized in detail using Easyfig (50) and Mauve (51).

**Whole-genome phylogenetic analysis of Stx2k-STEC strains.** Genome assemblies of 170 Stx2k-STEC strains in this study, together with other reported Stx2k-STEC strains (12, 14), were assessed by whole-genome multilocus typing (wgMLST) and whole-genome phylogeny. The complete whole-genome sequence of O157:H7 strain Sakai (NC_002695.2) was used as a reference to perform an *ad hoc* fast-GeP (Fast Genome Profiler) analysis (https://github.com/jizhang-nz/fast-GeP) to define wgMLST allelic profiles (52). Whole-genome phylogeny was inferred from the coding sequences (CDSs) shared by all genomes using Gubbins version 2.3.4 with the default settings (53). Single-nucleotide polymorphism (SNP) distances were executed using snp-dists version 0.7.0 (https://github.com/tseemann/snp-dists); the input alignment files were calculated by using snippy version 4.3.6 (https://github.com/tseemann/snippy) with the default settings and Gubbins version 2.3.4. The online tool ChiPlot (https://www.chiplot.online/#Phylogenetic-Tree) was used to visualize and annotate the phylogenetic tree.

**Data availability.** The assembled genomes of Stx2k-STEC strains in this study were uploaded to the NCBI GenBank database with the accession numbers SAMN27609646 to SAMN27609693, SAMN27609695 to SAMN27609767, and SAMN27609769 to SAMN27609814. The accession numbers for previously reported Stx2k-STEC strains used to determine phylogenetic relationships are SAMN12214771, SAMN24967101, SAMN24967107 to SAMN24967109, SAMN24967114, SAMN24967081, SAMN24967088, SAMN24967089, SAMN24967093, SAMN24967096, SAMN12214764, SAMN12214770, SAMN15722310, SAMN12214765 to SAMN12214767, SAMN12214763, SAMN12214768, SAMN12214769, SAMN27609646 to SAMN27609693, SAMN27609695 to SAMN27609767, and SAMN27609769 to SAMN27609814.

## SUPPLEMENTAL MATERIAL

Supplemental material is available online only.
**SUPPLEMENTAL FILE 1**, XLSX file, 0.1 MB.
**SUPPLEMENTAL FILE 2**, PDF file, 2.8 MB.

## ACKNOWLEDGMENTS

This study was supported by grants from the National Natural Science Foundation of China (grant number 82072254) and the National Science and Technology major project (grant number 2018ZX10301407-002).

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
