## [Reviewer comments · Microbiology Spectrum]

Microbiology Spectrum

High prevalence and persistence of *Escherichia coli* strains producing Shiga toxin 2k subtype in goat herds

Yanwen Xiong, Xi Yang, Qian Liu, Xiangning Bai, Bin Hu, Deshui Jiang, Hongbo Jiao, Liangmei Lu, Ruyue Fan, Peibin Hou, and Andreas Matussek

Corresponding Author(s): Yanwen Xiong, State Key Laboratory for Infectious Disease Prevention and Control, National Institute for Communicable Disease Control and Prevention, China CDC

Review Timeline:

Submission Date:	April 29, 2022
Editorial Decision:	May 29, 2022
Revision Received:	June 8, 2022
Accepted:	July 18, 2022

Editor: Sadjia Bekal

Reviewer(s): Disclosure of reviewer identity is with reference to reviewer comments included in decision letter(s). The following individuals involved in review of your submission have agreed to reveal their identity: Kenichi Lee (Reviewer #2)

Transaction Report:

DOI: <https://doi.org/10.1128/spectrum.01571-22>

May 29, 2022

Prof. Yanwen Xiong
State Key Laboratory for Infectious Disease Prevention and Control, National Institute for Communicable Disease Control and Prevention, China CDC
P.O. Box 5, Changping
Beijing 102206
China

Re: Spectrum01571-22 (High prevalence and persistence of Escherichia coli strains producing Shiga toxin 2k subtype in goat herds)

Dear Prof. Yanwen Xiong:

Link Not Available

Sincerely,

Sadjia Bekal

Journals Department
Reviewer comments:

Reviewer #1 (Comments for the Author):

This is a well written paper and was an interesting and informative read. The manuscript is built on a solid scientific foundation and includes a thorough sequence analysis of selected strains. The large scale sample collection survey conducted in this study is a huge undertaking and deserves much praise. The prevalence data is an important contribution to the research community as there is limited data available in the literature which investigates the prevalence of STEC within goats. The subsequent identification of a novel Stx2k subtype is an interesting and novel discovery. The authors must be commended for their thorough evaluation of the genomic architecture of the Stx2k prophage and the phylogenetic comparisons carried out, particularly with

Stx2k subtypes of clinical origin. The accompanying figures and supplementary figures are informative and very well designed.

Reviewer #2 (Comments for the Author):

Comments

In this study, authors found that recently found stx subtype, stx2k, can be isolated from goat at high rate. Whole-genome analyses of the isolates show the characteristics of the isolates (a few adhesins and other virulence factors, but half of them are positive for LT gene).

General comments

1. It is novel and informative that authors found that goat carry stx2k-positive E. coli at high rate, and characteristics of stx2k-positive isolates and Stx2k phage.
2. There are some technical concerns and points that should be clarified.

Specific comments

Line 27. It is better to mention LT gene (elt) rather than "virulence determinant."

Line 40. Highly homogeneous phage sequences suggest they are transmitting horizontally rather than vertically.

Line 114. It is important to mention that major adhesins of STEC (T3SS) and ETEC (e.g., F4, F5, F18) are absent.

Line 114. astA is very short gene and has high similarity with transposase gene. Therefore, usual homology search (e.g., 60% length match and 90% similarity match, VirulenceFinder default setting) often generates false positive. Please check whole coding region and start codon are conserved in this isolates.

Line 124. In Table S2, there are genes that are not directly gain antimicrobial resistance (e.g., H-NS). Authors should revise the genes.

Line 131. What about the similarities to the other stx2 phages?

Line 135. It is unclear how stx2 phages were clustered. Please show the results if possible.

Line 169. What are the criteria for clusters?

Line 178. It is difficult to say isolates in Fig S3d are clustered. There are thousands of SNPs.

Line 192. Can stx2k gene amplified by previous primers for stx2.

Line 237. The results suggest that horizontal gene transfer.

Line 269. It is not mentioned in the Results section.

Line 288. How many farms were included in this study? Were multiple isolates used in this study? If so, please include farm ID in Table S1.

Line 294. I suppose the original reference for stx-PCR is Bai et al. (2013, PLOS ONE, doi.org/10.1371/journal.pone.0065537). However, it is not mentioned which primers were used. The primer selection is important for this study and thus they should be mentioned clearly.

Line 307. Probably 250 bp would be correct.

Line 311. What was the criteria for PacBio sequencing? It would be helpful to show PacBio-sequenced isolates in Fig. 3.

Line 353. Were insertion sites detected by draft genomes?

Line 334. Were novel O-types that DebRoy et al. (PLoS One 2016 Vol. 11 Issue 1 Pages e0147434) proposed included?

Is it possible to measure the amount of Stx2k by commercial kits?

Staff Comments:

Preparing Revision Guidelines

For complete guidelines on revision requirements, please see the journal Submission and Review Process requirements at <https://journals.asm.org/journal/Spectrum/submission-review-process>. **Submissions of a paper that does not conform to**

Microbiology Spectrum guidelines will delay acceptance of your manuscript. "

Please return the manuscript within 60 days; if you cannot complete the modification within this time period, please contact me. If you do not wish to modify the manuscript and prefer to submit it to another journal, please notify me of your decision immediately so that the manuscript may be formally withdrawn from consideration by Microbiology Spectrum.

1 **High prevalence and persistence of *Escherichia coli* strains producing Shiga** 2 **toxin 2k subtype in goat herds**

**Xi Yang^a, Qian Liu^a, Xiangning Bai^{a,b,c}, Bin Hu^d, Deshui Jiang^e, Hongbo Jiao^e,**
**Liangmei Lu^e, Ruyue Fan^d, Peibin Hou^d, Andreas Matussek^{b,c,f}, Yanwen Xiong^{a,*}**

7 ^a State Key Laboratory of Infectious Disease Prevention and Control, National Institute for Communicable
Disease Control and Prevention, Chinese Center for Disease Control and Prevention, Beijing, China

9 ^b Division of Clinical Microbiology, Department of Laboratory Medicine, Karolinska Institute, Stockholm,
Sweden

11 ^c Division of Laboratory Medicine, Oslo University Hospital, Oslo, Norway

12 ^d Shandong Center for Disease Control and Prevention, Jinan, Shandong, China

13 ^e Lanling Center for Disease Control and Prevention, Lanling, Shandong, China

14 ^f Division of Laboratory Medicine, Institute of Clinical Medicine, University of Oslo, Oslo, Norway

**Abstract**

Shiga toxin (Stx)-producing *Escherichia coli* (STEC) is a zoonotic pathogen with the ability
to cause severe disease such as hemorrhagic colitis (HC) and hemolytic uremic syndrome
(HUS). Shiga toxin (Stx) is the key virulence factor in STEC, which can be classified into
two types, Stx1 and Stx2, and different subtypes. Stx2k is a newly-reported Stx2 subtype in
*E. coli* strains from diarrheal patients, animals, and raw meats exclusively in China so far. To
understand the reservoir of Stx2k-producing *E. coli* (Stx2k-STEC), we investigated
Stx2k-STEC in goat herds and characterized their genetic characteristics using
whole-genome sequencing. A total of 448 STEC strains were recovered from 2,896 goat
fecal samples and 37.95% (170/448) were Stx2k-STEC. Stx2k-STEC strains of O93:H28
serotype and ST4038 sequence type were the most predominant and detected over years.
Notably, 55% of Stx2k-STECs carried virulence determinant defining enterotoxigenic *E. coli*
(ETEC), thereby exhibiting the hybrid STEC/ETEC pathotype. Stx2k-converting prophage
genomes trended to cluster into four groups and exhibited high similarity within each group.
Strains from patients, raw meats, sheep and goat were mix-distributed in the phylogenetic
tree, indicating the risk for spread of Stx2k-STEC cross-species and pathogenic potential to

humans. Further studies are required to investigate the Stx2k-STECs in other reservoirs and
to understand the mechanism of persistence in these hosts.

**Importance**

The recently reported Stx2k-STEC strains are circulating in a variety of sources over time in
China. Here, we depicted a high prevalence of Stx2k-STEC in goat herds. More than half of
the strains were hybrid STEC/ETEC pathotype. Stx2k-STECs of predominant serotypes
were widespread in the goat herds over years. Stx2k-converting prophages exhibit a high
level of similarity across geographical regions and time, and might be maintained and
inherited vertically. The high prevalence of Stx2k-STEC in the goat suggests the spread risk
cross-species, given that goat-derived Stx2k-STECs share similar genetic backbones as
patients-derived strains, these strains may pose pathogenetic potential to humans. Our study
thus highlights the need of monitoring human Stx2k-STEC infections in this region, by
extension, in other geographic locations.

**Keywords** *Escherichia coli*, Shiga toxin (Stx), Stx2k subtype, Stx2k-converting prophage,
Whole-genome sequencing

**Introduction**

Shiga toxin-producing *Escherichia coli* (STEC) is a significant zoonotic foodborne pathogen
worldwide. The clinical manifestations of STEC infection range from asymptomatic carriage,
diarrhea or hemorrhagic colitis (HC) to the potentially life-threatening complication
hemolytic uremic syndrome (HUS), which could lead to acute renal failure (1). Ruminants
are important reservoirs of STEC, although cattle have been recognized as the primary
source of STEC, recent epidemiological studies have suggested that goats and sheep are also
significant sources of human STEC infection (2-4). Human STEC infection occur through
direct or indirect contact with animals, e.g., animal slaughter process or exposing to
feces-contaminated water, soils, animal-derived food and agricultural products.

STEC pathogenicity is mainly due to the production of one or more kinds of Shiga toxins
(Stxs), which is encoded in the lysogenized lambdoid prophages late region. Stx-converting
prophages, as horizontal gene transfer (HGT) elements, can convert a harmless commensal
into an enteric pathogen and promote the emergence of hybrid pathotypes (5), thereby
playing an important role in STEC pathogenesis and evolution. Stx is an AB₅ toxin and
includes two antigenic forms, Stx1 and Stx2. According to the nomenclature proposed by
Scheutz et al., several subtypes of Stx1 (Stx1a, Stx1c and Stx1d) and Stx2 (Stx2a to Stx2g)
have been described (6). Based on epidemiological studies, different Stx subtypes vary in
toxicity leading to different patient outcomes (7). Stx2 producing strains are associated with
more severe disease than Stx1 producing strains, particularly strains harboring Stx2a and
Stx2d subtypes (8). In recent years, new Stx2 subtypes have been identified: Stx2h to Stx2m
and Stx2o (8-13). Of these, Stx2j, Stx2k, Stx2m and Stx2o subtypes have been identified in
clinical isolates (11, 13), suggesting the clinical relevance of new Stx subtypes. Interestingly,
Stx2k-STECs have been identified in *E. coli* strains from a variety of hosts including
diarrheal patients, animals (goat, sheep and pig) and raw meat (mutton and beef) in China
over time(12, 14), while not yet been reported in any other countries.

In this study, we investigated the Stx2k-STECs in goat herd as one important reservoir
through our continuous STEC monitoring over years, whole-genome sequencing (WGS) was

used to characterize Stx2k-STEC strains, including their molecular traits, Stx2k-converting
phages, and genetic relatedness of strains from diverse sources.

**Results**

**High Prevalence of Stx2k-STEC Strains in Goat Herds**

In total, 448 STEC strains were isolated from 2,896 goat fecal samples, giving a culture
positive rate of 15.47% (448/2896). The isolation rate of STEC from goat feces during the
study period is shown in Table 1. A total of 12 different *stx* subtypes/combinations were
identified among 448 STEC isolates, with *stx2k* (169/448, 37.72%) being the most
predominant, followed by *stx1c* (120/448, 26.69%) and *stx1c+stx2b* (65/448 14.51%). One
isolate carried *stx2k* and *stx2e*. The proportion of *stx2k*-positive strains was highest in 2019
and 2021 (Figure 1).

Among the 170 *stx2k*-positive STEC strains, 167 shared identical *stx2k* sequence as the
previously reported Stx2k-STEC strain STEC309 (GenBank: CP041435.1) (12), the
remaining three strains harbored the same *stx2k* sequence which differed from strain
STEC309 at position 346 in the A subunit, leading to a change in one amino acid (from D to
97 N). The B subunit and intergenic region between A and B subunit (aggagttaagt) were highly
conserved among all goat-derived Stx2k-STECs in this study.

**Molecular Characteristics of Goat-derived Stx2k-STEC Isolates**

Among the 170 goat-derived Stx2k-STEC strains, 16 different O:H serotypes were found.
O93:H28 was the most predominant serotype accounting for 30.59% (52/170) of all strains
(Table S1). Two novel O-serogroup genotypes (OgN-RKI3:H21 and OgN17:H21) were
found among 34 Stx2k-STEC isolates (20.0%) (Table 2). Sixteen MLST sequence types
(STs) were assigned among all isolates, of these, ST4038 was most predominant (52/170,
30.59%). Isolates with the same serotype were assigned as the same sequence type, with the
exception of OgN-RKI3:H21 (ST683 and ST58) and O116:H25 (ST155 and ST58) (Table 2).
The dominant serotype (sequence type) in 2017 and 2019 was O93:H28 (ST4038), whereas
in 2018 and 2021, OgN-RKI3:H2 (ST683) predominated.

A total of 95 virulence genes were identified among 170 Stx2k-STEC strains based on the
VFDB database. These virulence genes can be classified into several groups based on their
functions, in addition to *stx*, the enteroaggregative *E. coli* heat-stable enterotoxin (EAST1)
encoding gene *astA* was observed in 17 strains. Ninety-four Stx2k-STEC strains (55.3%) in
this study carried heat-labile toxin (LT) encoding gene, a virulence determinant for
enterotoxigenic *E. coli* (ETEC), thereby exhibiting the hybrid STEC/ETEC pathotype. All of
the STEC/ETEC strains carried the heat-labile toxin-encoding gene *lt-II* (Table S1). Other
virulence genes were involved in adherence (*fdeC*, *cfaABCDE*, *cgsABCDEFG*,
*fimABCDEFGH* and *ecpABCDER*), invasion (*aslA*, *ibeB*, *ompA* and *AAA92657*), iron
uptake (*ybtS*, *fepABCDEG* and *fur*), secretion system effectors (*yopP*, *espLRXY*, *fha rhs*,
*tssABCDEFGHIJKLM* and *gspCDE-KLM*), regulation (*phoP*, *rcsB* and *rpoS*), etc. (Table
S2a).

Sixty-seven AMR genes were identified, 42 of them were efflux-related transporter
protein-encoding genes (Table S2b). AMR genes associated with resistance to peptide
antibiotics (*pmrF*) and class C β -lactamase (*bla_{EC}*, *ampH* and *ampC1*) were found in all
strains. Other predominant AMR genes were *tet(B)* (n=29), *aph(6)-Id* (n=17), *qnrS1* (n=16),
*bla_{TEM-181}* (n=16) and *sul3* (n=16).

**Genetic Feature of Stx2k-Converting Prophages**

Twelve intact and 158 incomplete Stx2k-converting prophages were identified from 170
Stx2k-STEC genomes. To assess the genetic relationships among all of the Stx2k-converting
prophages in this study and reference Stx2k-STECs reported previously, intergenomic
similarities were calculated reciprocally between pairs of prophage genomes using VIRIDIC,
four Stx2k prophage groups, termed as Stx2k-prophage G1, G2, G3 and G4, were designated
based on the cluster assignment. Stx2k prophages in this study (n=170) were distributed in
G1 to G3, and one Stx2k prophage from a reference pig-derived STEC strain was grouped as
G4 (Figure S1). Prophages identified from draft genomes with length $\leq 8,000$ bp were
assigned as unknown group (Gx). Three different integrases were identified in 44 prophages
(data not shown), consistently, three insertion sites were found, i.e., *yccA* (Modulator of FtsH
protease YccA), *dusA* (tRNA dihydroxyuridine synthase A), and *yecE* (DUF72

domain-containing protein YecE) (Table S1). The integrases and insertion sites corresponded
to the three Stx2k-prophage groups (Figure 2).

To characterize and compare the Stx2k-converting prophages from strains of different
molecular characteristics, Stx2k-STEC strains representative of different serotypes and
sequence types were selected for further comparisons. The progressive Mauve alignment
revealed the blocks with collinearity between Stx2k-converting prophages (Figure S2). Stx2k
prophages of the same group displayed nearly identical collinearity blocks component.
Consistent results were observed using Easyfig. The Stx2k-converting prophages were
generally organized into three major modules, i.e., integration cassette, lysis cassette
functions and morphogenetic related genes. The Stx2k-converting prophage aligned across
the length of the prophage with few structural variations in each group. For instance, the
insertion sites, integrase, replication proteins and the antiterminator protein Q of the phages
(Figure 2). All of the Stx2k prophages shared a conserved region surrounding *stx* of
approximately 40 kb. The prophage structure among those groups mainly differed in their
early regions (upstream of *stx2k*), especially the genes encoding regulator and replication,
while only minor structural variations were observed in their late regions coding the
morphogenetic and lysis cassette functions.

**Phylogenetic analysis**

To determine the phylogenetic relationships of Stx2k-STEC strains isolated from goat in this
study and 17 previously reported Stx2k-STEC strains from diverse sources (12, 14), a
whole-genome phylogeny tree was constructed from alignment of concatenated CDSs of the
1,909 shared-loci found in all 187 Stx2k-STEC isolates (Figure 3). Strains isolated from
different years were inner-mixed in the phylogenetic tree. Strains belonging to the same
serotype and MLST type showed the tendency to cluster together. Pairwise SNP distance
heatmaps were produced to illustrate the dissimilarity of strains within each cluster (Figure
S3). Two large clusters were observed comprising of 36% and 31% isolates respectively,
referred to as cluster 1 and cluster 2. Cluster 1 primarily consisted of 47 OgN-RKI3:H21
isolates from goat, sheep and patient from three different provinces, patient-derived strains
and other isolates differed in 292 to 720 SNPs (Figure S3a). All isolates from the cluster 2

belonged to O93:H28 isolated from goat in this study. Besides cluster 1, four clusters
possessed strains isolated from different sources or geographical locations were observed.
All isolates in cluster 3 belonged to O100:H19 serotype isolated from goat and raw meat,
strains from the two sources differed in 62 to 265 SNPs (Figure S3b). Goat-derived STECs
isolated from two provinces were grouped in cluster 4, which differed in 458 to 459 SNPs
(Figure S3c). Cluster 5 consisted of one patient-derived isolate and goat isolates belonging to
different serotypes, the differences between strains from the two sources ranged from 8,915
to 9,984 SNPs (Figure S3d). One pig-derived O159:H16 STEC isolate formed a separate
cluster. Strains of same phylogenetic cluster carried Stx2k-phages assigned as different
phage groups (Figure 3).

**Discussion**

Stx2k-producing *E. coli* strains have been identified from a broad range of hosts exclusively
in China so far, including diarrheal patients, pig, raw meat (beef and mutton), sheep and
goat(12, 14). In this study, 170 Stx2k-STEC strains were isolated from goat and the
proportion of Stx2k (37.95%) was the highest among all *stx* subtypes detected. Goat have
been identified as significant reservoirs of STEC and are capable to carry STEC without
clinical signs themselves (3, 4). Previous studies have suggested that the most frequent *stx*
type encountered in goat-derived isolates was *stxI* (15-17) and especially *stxIc* subtype (18,
19). This might be due to the lack of systematic investigation of STEC among goat (4),
besides, since the application of WGS becomes routine for STEC genotyping, the diversity
of *stx* subtypes in STEC from different sources is accumulating. Stx2k-STECs have only
been reported in China so far, it is not unlikely that Stx2k-STEC may occur in other
countries but not identified or assigned as other Stx2 subtype, for instance, Stx2k was
initially assigned as a new Stx2e variant (20). One example is that the recently designated
Stx2h subtype has previously only been detected in isolates from Tibetan marmots (9),
however, it has now also been reported recently from sprouts in Canada (11). The host range
and specificity of different *stx* subtypes need further study.

Previous studies have reported differences between Stx1/Stx2 types and subtypes in terms of
host specificity and binding strength (21, 22). One Stx2k variant was reported less toxic in

Vero cell assay than Stx2a but similar in receptor-binding preference, acid tolerance, and
thermostability (23). It has been proposed that the two amino acid differences in the receptor
binding site might explain difference in cytotoxicity between Stx2a and Stx2k. The similarity
between Stx2k and Stx2a, and the high prevalence of Stx2k-STEC in goat herds suggest a
possible risk to cause human infections. In addition to three *stx2k* variants previously
reported (12), a new *stx2k* variant was identified in this study, thus there are four variants in
Stx2k-STEC to date. It remains to be investigated whether the amino acid change between
different variants in the catalytically active A-subunit plays a role in the toxicity of Stx2k.

The WGS-based molecular characterization revealed that O93:H28 was the most
predominant serotype in goat-derived Stx2k-STEC strains. To the best of our knowledge, no
human disease has been reported to be associated with this serotype thus far. The reason for
this may be that most of the serotyping is performed by conventional slide agglutination tests
using antisera against the recognized *E. coli* O antigens established by the International
Centre for Reference and Research on *Escherichia* and *Klebsiella*, and the former scheme
did not include O93 (24, 25). The novel O-serogroup genotypes OgN-RKI3 has exclusively
been reported in human-derived STEC strain 16-04178 (GCA_016116345.1) (26, 27),
carrying *stx2e* subtype, our finding that 18% goat-derived Stx2k-STEC strains in this study
possessed the same O genotype as strain 16-04178, indicating that OgN-RKI3 strains have a
potentially wide distribution. Remarkably, 55% of Stx2k-STEC strains in our study carried
the heat-labile toxin (LT)-encoding gene *lt*, exhibiting a STEC/ETEC hybrid pathotype. In a
recent study, 56% of Stx2e-STEC strains exhibited the STEC/ETEC hybrid pathotype (28).
Notably Stx2k was initially assigned as a new Stx2e variant from its type strain STEC388
isolated from a healthy pig (20). Stx2k- and Stx2e-producing STEC strains may share similar
genetic backbones that can drive the transmission of genes among strains, facilitating the
emergence of hybrid pathotypes, further research is needed to elucidate this.

Clustering of Stx2k-converting prophages demonstrates the similarity among Stx2k-
prophages. This was consistent with a previous study indicating that Stx2a phages exhibited
overall sequence similarities to each other (29), and that the Stx2c phages of the O157 strains
were highly homogeneous (30, 31). In contrast Stx2e prophages exhibits a considerable

diversity (28, 32). In this study, Stx2k-converting phages from diverse sources or
geographical regions were distributed in three designed prophage groups. This might be due
to vertically maintain and inherit of the Stx2k-converting prophages. Comparing of
Stx2k-converting prophages in terms of genomic organization and nucleotide sequence of
homologous genes highlighted the similarity within each group. Stx phage integrases
determined the reorganization of different insertion sites within the bacterial chromosome (5).
Among the three insertion sites in the goat-derived Stx2k-STECs, *yecE* was commonly
found in *stx* prophages (28, 32-34), whereas the other two insertion sites were rarely reported.
*yccA*, associated with the Sp4-prophage in O157:H7 (30), was found as insertion site in *stx2e*,
*stx2k* and *stx2a*-converting prophages (12, 28, 35). Insertion site was usually located
upstream of the phage, in the predominate group G1, insertion site *dusA* was located
downstream of the phage and was found in *stx2k*, *stx2a* and *stx2d*-converting Stx prophages
(12, 35). This tRNA-dihydrouridine synthase also carries a phage scar in O157:H7 (30). Our
study showed that the genomic diversity between each Stx2k phage group mainly occur at
their early regions that contained regulator and replication related genes. This region
mediates the switch between repression and induction of the prophage (36), thus may
influence the expression level of the Shiga toxin (29) as well as the potential to produce
phage particles without inducer agents of STEC strains (37). The difference between Stx2k
prophage groups and the presence of conserved regions highlighted the modular structure of
lambdoid phages (38).

The whole-genome phylogeny of all goat-derived Stx2k-STECs in this study demonstrated a
considerable genetic diversity. The dominant clusters (Cluster 1: OgN-RKI3:H21, ST6830
and Cluster 2: O93:H28, ST4038) were widespread in this region in the goat herds over
259 years. When compared with other reported Stx2k-STECs from different sources, a high
degree of genetic similarity was observed between Stx2k-STECs from goat and other sources.
Here patient-derived and goat-derived strains clustered phylogenetically closely, suggesting
the pathogenic potential of goat-derived Stx2k-STECs. Further studies are essential to
investigate the prevalence of Stx2k-STEC among humans in this region, and to assess their
pathogenic potentials. The genomic similarity between aforementioned Stx2k-STEC isolates,
despite their timing of isolation, geographic differences, and origin of isolation, may indicate

the widespread of these strains cross-species and in different regions in China. Further
studies are required to investigate Stx2k-STEC in diverse sources especially humans, and in
other georgical regions in China and abroad. Furthermore, the distribution of Stx2k prophage
groups in the phylogeny tree demonstrates that the Stx2k prophage does not parallel the
whole-genome phylogeny, suggesting that Stx2k phage and its bacterial host evolve
independently. Strains in the same clusters might descend from one ancestor and acquire the
Stx2k prophages in independent transduction events.

In conclusion, this study reports a high prevalence of the newly-reported Stx2k subtype in
STEC strains in goat herds. Strains of predominant serotypes and Stx2k-converting prophage
groups were recovered over years. The goat-derived Stx2k-STEC strains in this study
demonstrated genetic similarity with those from other sources. Thus, possible transmission
and pathogenic potential of Stx2k-STEC to humans should be noted.

**Materials and Methods**

**Ethics statement**

The current study has been reviewed and approved by the ethic committee of the National
Institute for Communicable Diseases Control and Prevention, China CDC, with the number
ICDC-2017006. Fecal samples of goat were acquired with the consent of the owners of
animals.

**Sample Collection and Isolation of STEC Strains**

In total, 2,896 fecal samples from healthy-looking goat were collected from different
family-scale farms in Lanling county, Shandong province, China, from November 2017 to
October 2021. Fecal samples were collected in 2 mL sterile tubes containing Luria-Bertani
(LB) medium in 30% glycerol, and transported to the laboratory in the National Institute for
Communicable Disease Control and Prevention, China CDC in ice-bags for the isolation of
STEC. Strains were isolated and confirmed by methods described previously (39). Briefly,
after enrichment with EC broth, the samples were examined by PCR for presence of *stx*.
Samples positive for *stx* were inoculated on CHROMagarTM ECC agar (CHROMagar, Paris,
France). After overnight incubation at 37°C, green-blue or colorless colonies on agar were

picked and tested for *stx* by a single colony duplex PCR assay (39). API 20E biochemical
test strips (bioMérieux, Lyon, France) were used to confirm that all *stx*-containing isolates
were *E. coli*.

**Whole Genome Sequencing (WGS) and Genome Assembly**

Bacterial DNA extraction was performed with the Wizard Genomic DNA purification kit
(Promega, Madison, WI, USA) according to the manufacturer's protocol. Library preparation
and WGS were performed at Beijing Novogene Bioinformatics Technology Co., Ltd., China.
To obtain the draft genomes of STEC, DNA library preparation was done using NEBNext[®]
Ultra[™] DNA Library Prep Kit (New England Biolabs, Ipswich, MA, USA). The library was
then paired-end (2 × 350 bp) sequenced using the Illumina NovaSeq platform (Illumina, San
Diego, CA, USA). The low-quality reads (quality score < Q20) were filtered by fastp 0.20.1
(<https://github.com/OpenGene/fastp>) and then *de novo* assembled using SKESA version
2.4.0 (<https://github.com/ncbi/SKESA>), low-quality contigs (length < 500 bp) were
filtered with Seqkit version 0.11.0. To obtain the complete genomes of selected Stx2k-STECS
strains, two sequencing libraries were prepared, besides the Illumina DNA library, a 10Kb
library was done using a SMRT bell Template Prep kit (version 1.0). The long library was
sequenced using PacBio Sequel platform (Pacific Biosciences, Menlo Park, CA, USA). The
long reads were preliminarily quality-controlled using "RUN QC" module in SMRT Link
version 5.1.0 (www.pacb.com/support/software-downloads), and *de novo* assembled using
the Hierarchical Genome Assembly Process (HGAP) pipeline, then corrected using the
Illumina short reads to get complete genomes. All genomes were annotated with Prokka
(version 1.13.3) (40).

**Molecular Characterization of STEC Isolates**

Molecular characterization was based on WGS data including *stx* subtyping, serotyping,
multi-locus sequence typing (MLST), detection of virulence factor genes and antimicrobial
resistance genes. Briefly, an in house *stx*_subtyping database including representative
nucleotide sequences of all identified *stx1* (*stx1a*, *stx1c*, and *stx1d*) and *stx2* (*stx2a* to *stx2m*
and *stx2o*) subtypes was created, assemblies were compared against the *stx*_subtyping
database using ABRicate version 0.8.10 (<https://github.com/tseemann/abricate>) to determine

the *stx* subtypes. Similarly, a serotyping database was created including all known *wzx/wzy*
sequences from typical *E. coli* O-serogroups from O1 to O188, *fliC* and its homologs
sequences from typical *E. coli* H types from H1 to H56, download from Center for Genomic
Epidemiology (CGE) (database version 1.0.0)
(https://bitbucket.org/genomicepidemiology/serotypefinder_db/src/master/)(41), and a set of
recently defined novel O-antigen biosynthesis gene cluster types (Og types) types of *E. coli*
(26, 42). MLST of seven housekeeping genes was performed through an on-line tool
provided by the Warwick *E. coli* MLST scheme website
(https://enterobase.warwick.ac.uk/species/ecoli/allele_st_search). Detection of virulence
genes and antimicrobial resistance genes was performed by comparing assemblies against
the virulence factor database (VFDB) (43), and the Comprehensive Antibiotic Resistance
Database (<http://arpcard.mcmaster.ca>), respectively, using ABRicate with default parameters
(coverage $\geq 80\%$ and identity $\geq 80\%$). Given that some Stx2k-STEC strains carried
heat-stable and heat-labile enterotoxin encoding genes (*st* and *lt*) according to a previous
study (12), we further subtyped *st* and *lt* genes by comparing assemblies against an *in-house*
database including representative nucleotide sequences of *st* and *lt*, using ABRicate version
0.8.10. Nucleotide sequences of different *st* and *lt* subtypes were obtained from previous
reports (44-46).

347 **Identification and Characterization of Stx2k-converting Prophages**

Stx2k prophage sequences were extracted from genomes and characterized using methods
previously described (28). Briefly, PHAge Search Tool Enhanced Release (PHASTER,
<http://phaster.ca/>) was used to identify Stx prophages, the intact and incomplete Stx2k
prophage sequences were extracted from complete and draft genomes, respectively. The
RAST server (<http://rast.nmpdr.org/>) was used to annotate the genomes of Stx prophages.
The phage insertion site was defined as the gene adjacent to the integrase (32). To assess the
genetic relationship of Stx2k prophages, intergenomic similarities between pairs of prophage
genomes was calculated using the VIRIDIC (Virus Intergenomic Distance Calculator) phage
genome comparison tool with default parameters (47). The sequences of Stx2k prophage
were compared and visualized in details using Easyfig (48) and Mauve (49).

**Whole-Genome Phylogenetic Analysis of Stx2k-STECs**

Genome assemblies of 170 Stx2k-STECs in this study together with other reported
Stx2k-STECs (12, 14) were assessed by whole-genome multilocus typing (wgMLST) and
whole-genome phylogeny. The complete whole-genome sequence of O157:H7 strain Sakai
(NC_002695.2) was used as a reference to perform an *ad hoc* fast-GeP analysis
(<https://github.com/jizhang-nz/fast-GeP>) to define wgMLST allelic profiles (50).
Whole-genome phylogeny was inferred from the coding sequences (CDSs) shared by all
genomes using Gubbins (version 2.3.4) with default settings (51). Single-nucleotide
polymorphisms (SNPs) distances were executed using snp-dists v0.7.0
(<https://github.com/tseemann/snp-dists>), the input align files were calculated by snippy
v4.3.6 (github.com/tseemann/snippy) with default settings and Gubbins (version 2.3.4). An

[revised manuscript text omitted]

- 501

502 **Table 1.** Prevalence of STEC in goat herd from 2017 to 2021.

Year	No. of samples	No. of STEC isolates (%^a)	No. of Stx2k-STECs (%^b)
2017	407	83 (20.39)	20 (24.10)
2018	325	32 (9.85)	12 (37.5)
2019	512	86 (16.80)	37 (43.02)
2020	646	108 (16.72)	36 (33.33)
2021	1006	139 (13.82)	65 (46.76)
Total	2896	448 (15.47)	170 (37.95)

503 ^a Culture-positive rate of STEC strains among all samples.

504 ^b Prevalence of Stx2k-STEC among all STEC isolates.

505

506 **Table 2.** Serotypes and sequence types of the 170 Stx2k-STEC isolates from 2017 to 2021.

Serotype	MLST (No. of isolates)	2017	2018	2019	2020	2021
O93:H28	ST4038 (52)	12	-	31	8	1
OgN-RKI3:H21	ST683 (30), ST58 (1)	-	10	1	2	18
O184:H19	ST6313 (17)	-	-	-	-	17
O174:H2	ST13029 (13)	-	-	-	-	13
O22:H8	ST446 (12)	3	-	1	3	5
O133:H25	ST10326 (11)	-	1	2	8	-
O16:H32	ST1286 (8)	-	-	-	8	-
O22:H16	ST295 (5)	-	-	-	-	5
O116:H25	ST155 (3), ST58 (1)	2	-	1	1	-
O8:H19	ST162 (4)	-	-	-	4	-
OgN17:H21	ST602 (3)	-	-	1	2	-
O100:H19	ST1611 (3)	-	-	-	-	3
O159:H25	ST155 (3)	-	-	-	-	3
O8:H53	ST345 (2)	2	-	-	-	-
O86:H51	ST155 (1)	1	-	-	-	-
O48:H21	ST5221 (1)	-	1	-	-	-
Total	170	20	12	37	36	65

**Figure Legends**

**Figure1.** Distribution of Shiga toxin subtypes from 2017 to 2021.

**Figure 2.** Easyfig plot comparing Stx2k-converting prophages from Stx2k-STEC strains
representative of different serotypes and sequence types within each prophage group. Arrows
indicate gene directions. Coding sequences are represented by arrows and linked by blue bars
shaded to represent the nucleotide identity, as indicated in the legend. The color of the text
indicates the source of strains, black represents goat-derived strains in this study and light
blue represents other source-derived strains from other studies. The asterisk (*) signifies
incomplete prophages. Prophages belonging to the same groups are shown in grey rectangle
and the Stx2k-prophage group is shown at the bottom-right corner.

**Figure 3.** Whole-genome phylogeny of Shiga toxin 2k-producing *E. coli* (Stx2k-STEC)
isolates. Circular representation of the Gubbins phylogenetic tree generated from the
concatenated sequences of the shared loci found in the wgMLST analysis. Gubbins tree was
annotated with relevant metadata using an online tool ChiPlot (<https://www.chiplot.online/>).
From the outer to inner circle, each represents serotype, year of isolation, phylogeny cluster,
Stx2k prophage group and isolation source. The leaves of the tree indicated MLST sequence
type.

**SUPPLEMENTARY MATERIALS**

**Table S1.** Molecular characteristics of Stx2k-STEC strains in this study and other sources.

**Table S2.** Virulence genes and antimicrobial resistance genes of the 170 Stx2k-STEC strains
in this study.

**Figure S1.** VIRIDIC comparisons of 187 *stx2k* prophages genomes. The comparison,
clustering and intergenomic similarity values between pairs of prophage genomes were
performed using Viridic. Online tool ChiPlot
(<https://www.chiplot.online/#Phylogenetic-Tree>) was used to visualize and annotate the
heatmap. The colored bar represents Stx2k prophage groups.

**Figure S2.** Sequence comparison of the Stx2k-converting prophages from Stx2k-STEC
strains representative of different serotypes and sequence types. Phage sequences extracted
from the genomes of the Stx2k-STEC isolates were subjected to sequence analysis using
Mauve. Similar color denotes regions of shared sequence, and the height of the bars denotes

level of similarity of the shared sequence regions. The color of the text indicates the source
of strains, black represents goat-derived strains in this study, and light blue represents other
source-derived strains in other studies. The asterisk (*) signifies incomplete prophages. The
location of the *stx* genes is shown with a red arrow. The colored bar in the right represents
the four prophage groups.

**Figure S3.** SNP differences of Stx2k-STEC strains within same phylogenetic clusters.
Heatmap illustrating pairwise SNP distances for (a) 62 isolates and reference strain
STEC309 (NZ_CP041435.1) in cluster 1, (b) 6 isolates and reference strain STEC367
(NZ_CP041429.1) in cluster 3, (c) 5 goat-derived isolates from different regions and
reference strain STEC005 (NZ_CP041437.1) in cluster 4, (d) 26 isolates and reference strain
STEC409 (NZ_CP041422.1) in cluster 5. The maximum likelihood trees presented as
cladograms, constructed using all isolates and reference strain in each cluster as mentioned
above. All trees were constructed using SNP alignments.

Dear Editor,

We sincerely thank editor and reviewers for critical assessments and valuable suggestions on the manuscript (Spectrum01571-22) entitled "High prevalence and persistence of *Escherichia coli* strains producing Shiga toxin 2k subtype in goat herds". We have revised the manuscript according to the reviewer's comments. We addressed reviewers' comments point by point as detailed below. We have uploaded manuscript with track changes in PDF file named 'Marked Up Manuscript-For Review Only' for comparison.

Reviewer #1 (Comments for the Author):

This is a well written paper and was an interesting and informative read. The manuscript is built on a solid scientific foundation and includes a thorough sequence analysis of selected strains. The large scale sample collection survey conducted in this study is a huge undertaking and deserves much praise. The prevalence data is an important contribution to the research community as there is limited data available in the literature which investigates the prevalence of STEC within goats. The subsequent identification of a novel Stx2k subtype is an interesting and novel discovery. The authors must be commended for their thorough evaluation of the genomic architecture of the Stx2k prophage and the phylogenetic comparisons carried out, particularly with Stx2k subtypes of clinical origin. The accompanying figures and supplementary figures are informative and very well designed.

We sincerely thank the reviewer's positive evaluation on our manuscript. We didn't find other specific comments from the reviewer (if any), the attachment is the original manuscript.

Reviewer #2 (Comments for the Author):

Comments

In this study, authors found that recently found *stx* subtype, *stx2k*, can be isolated from goat at high rate. Whole-genome analyses of the isolates show the characteristics of the isolates (a few adhesins and other virulence factors, but half of them are positive for LT gene).

General comments

1. It is novel and informative that authors found that goat carry *stx2k*-positive *E. coli* at high rate, and characteristics of *stx2k*-positive isolates and Stx2k phage.
2. There are some technical concerns and points that should be clarified.

Specific comments

Line 27. It is better to mention LT gene (*elt*) rather than "virulence determinant."

We changed this as suggested (line 29).

(Line number in this letter correspond to that shown in PDF file 'Marked Up Manuscript-For Review Only', which differ from clean .doc file 'Revised manuscript'.)

Line 40. Highly homogeneous phage sequences suggest they are transmitting horizontally rather than vertically.

Thanks for the correction. We have checked and corrected this throughout the manuscript.

Line 114. It is important to mention that major adhesins of STEC (T3SS) and ETEC (e.g., F4, F5, F18) are absent.

We have mentioned that the major adhesin of STEC (T3SS) (e.g., intimin encoded by *eae* on the locus of enterocyte effacement (LEE) pathogenicity island) and ETEC (e.g., F4, F18, F5, and F6) are absent in Result and Discussion sections as suggested (line 132, line 267).

Line 114. *astA* is very short gene and has high similarity with transposase gene. Therefore, usual homology search (e.g., 60% length match and 90% similarity match, VirulenceFinder default setting) often generates false positive. Please check whole coding region and start codon are conserved in this isolates.

We sincerely thank for the reviewer's critical comments. We screened the virulence factor genes using ABRicate with default parameters (coverage \geq 80% and identity \geq 80%) against reference sequences downloaded from VirulenceFinder database, 17 identical sequences (114 bp) showed closest nucleotides sequences identity (97.32%) and coverage (95.73%) with *astA*.

As suggested, we tried to extract the whole coding region and start codon of these proposed *astA* genes, but failed. We observed that the start codon in the reference *astA* sequence (GenBank: L11241.1) were missing in the 17 sequences in this study, see result (Figure 1) attached below. As the reviewer mentioned, partial coding sequence of IS256-like element IS1414 family transposase (GenBank: WP_000343747.1) were similar with *astA* (Figure 1), while their reading frames were different. The sequence in this study (114 bp)

was identical to the partial sequence of IS1414 family transposase, but a complete coding sequence of the transposase was not identified among genomes in this study.

Figure 1. The alignment of proposed *astA* sequences in this study, whole coding region of reference nucleotide sequence of *astA* and partial coding sequence of IS1414 family transposase.

We have deleted relative information concerning *astA* from manuscript and Table S2. It should be noted that this is a sort of classic problem when screening genes spectrum based on WGS data with a fixed cut-off value of sequence identify/coverage, one should manually check genes of particular interest, genes that have high similarity with other genes, etc. We apologize for our neglect and thanks again for this critical comment.

Line 124. In Table S2, there are genes that are not directly gain antimicrobial resistance (e.g., H-NS). Authors should revise the genes.

Thanks for the suggestion. We checked the genes and moved *bacA* and *eptA* to the class of peptide antibiotics, other genes that are not directly gain antimicrobial resistance, e.g., efflux pump were deleted, we revised text in the manuscript accordingly (Results, Table S2).

Line 131. What about the similarities to the other Stx2 phages?

In our previous study (Yang et al. Int J Med Microbiol. 2020 Jan;310(1):151377.) we have compared nine Stx2k-converting prophages with eight reference Stx2 subtype a – h, the results showed that different Stx2 phages (with different Stx2 subtypes) had diverse insertion sites and extensive non-homologous regions (Figure 2 from paper 2020).

Figure 2. Genome alignment of the nine Stx2k prophages and representative prophages carrying different *stx2* subtypes. Homologous regions between the genomes are represented with blocks of identical colors.

Due to the extensive non-homologous regions of different Stx2 phages, it might not make much sense to compare sequence similarity between different Stx2 phages as a whole, especially in the current situation that a few Stx2 phages genomes are intact, while most are not. In terms of structure similarity, as we mentioned above, we observed extensive non-homologous regions in different Stx2 phage. In addition, we have now compared each Stx2k prophage group (G1-G4) in this study with other Stx2 phage that exhibited similar structure to a specific Stx2k phage group (Figure 3). We observed structure difference in the phages that are more similar compared to other Stx2 phages. However, the reference Stx2 phage selected for comparison could affect the result, since Stx2 phages carrying the same Stx2 subtype may be diverse, as we reported previously (Yang et al. Pathogens. 2021 Nov 28;10(12):1551). As part of our ongoing work, we will try to induce different Stx2 phages to get complete genomes of different Stx2 subtype phages and explore this question in detail. We hope to share these data soon.

Figure 3. Easyfig plot comparing of Stx2k-converting prophages from each prophage group with the similar Stx2 prophages. Arrows indicate gene directions. Coding sequences are represented by arrows and linked by grey bars shaded to represent the nucleotide identity, as indicated in the legend. Red arrows indicate Shiga toxin. The color of the text indicates the Stx subtype of strains, red represents Stx2k-STEC strains in this study and blue represents Stx2e-STEC strains, green is Stx2d and yellow is Stx2a. The GenBank accession numbers are listed in bracket.

Line 135. It is unclear how stx2 phages were clustered. Please show the results if possible.

The clustering of Stx2k prophage genomes was calculated using the VIRIDIC (Virus Intergenomic Distance Calculator). Briefly, the clustering was based on genomic similarity of prophages. First, each prophage was aligned against all other genomes in the dataset, using BLASTN 2.9.0+ with the core parameters “-evalue 1 -max_target_seqs 10,000 -num_threads 6”, the default alignment parameters were “-word_size 7 -reward 2 -penalty -3 -gapopen 5 -gapextend 2”. Second, the BLASTN output was used to calculate pairwise intergenomic similarities. Third, VIRIDIC performed a hierarchical clustering of the intergenomic similarity values by using the fastcluster v. 1.1.25 R package (<https://cran.r-project.org/web/packages/fastcluster/index.html>). Fourth, VIRIDIC clustered and graphically represented the intergenomic similarity values as a heatmap visualization (see Figure 4a), using the ComplexHeatmap v.

2.5.3 R package. Since prophages extracted from draft genomes were incomplete, character of prophage structure (e.g., downstream or upstream gene) should also be considered when analyzing similarity among Stx2k prophages. Stx2k-converting prophages clusters sharing identical downstream or upstream gene were recognized as the same group (Figure 4b). We modified the method and result in line 417 and 155, and Figure S1.

Figure 4. (a). VIRIDIC heatmap incorporating intergenomic similarity values (right half) and alignment indicators (left half and top annotation) of Stx2k-converting prophages. Colored rectangle indicates hierarchical clusters. (b). Heatmap of pairwise intergenomic distances of 187 Stx2k-converting prophages assigned as different phage groups. The comparison, clustering and intergenomic similarity values between pairs of prophage genomes were performed using VIRIDIC. Online tool ChiPlot (<https://www.chiplot.online/#Phylogenetic-Tree>) was used to visualize and annotate the heatmap. The colored bar represents Stx2k prophage groups.

Line 169. What are the criteria for clusters?

Based on the whole genome phylogenetic tree, six phylogenetic clusters were designed for further comparison. The purpose is to understand the genetic difference of strains with similar genetic background but from different sources or geographical locations. We modified this in line 194.

Line 178. It is difficult to say isolates in Fig S3d are clustered. There are thousands of SNPs.

We understand the reviewer's concern. As mentioned above, the clusters were defined more from a phylogenetic point of view, when isolates 1) were phylogenetically grouped based on whole genome phylogeny; and 2) were isolated from different sources or geographical locations, the purpose was to compare the genetic difference of the isolates with similar genetic background but from different sources or locations. Strains within cluster 5 were indeed more diversely distributed compared to other clusters, if we have used a stricter cut-off value based on sequences similarity (e.g., SNP difference), cluster 5 could be divided into different sub-clusters as the reviewer mention, however, this is not our ambition, we have clarified this in the manuscript (line 205).

Line 192. Can *stx2k* gene amplified by previous primers for *stx2*.

Yes, *stx2k* gene can be amplified by our previous primers for *stx2*. We also have mentioned in our previous study (Yang, Bai et al.2020) "*stx2k* subtype is covered by the screening PCR recommended by the EU reference lab for *stx2* detection (<http://old.iss.it/vtec/index.php?lang=2&tipo=3>) with one mismatch in the forward primer (Supplementary Fig. S2)".

Line 237. The results suggest that horizontal gene transfer.

We have corrected it.

Line 269. It is not mentioned in the Results section.

We mentioned it in the result under subtitle Phylogenetic analysis: Different *Stx2k* prophage groups were observed within the same phylogenetic cluster (line 208-209).

Line 288. How many farms were included in this study? Were multiple isolates used in this study? If so, please include farm ID in Table S1.

We collected 2,896 samples in Lanling county from 2017 to 2021. For 2164 samples collected from 2019 to 2021, we collected detailed information including the family-scale farms (n=121), the sampling numbers

ranged from 1 to 122 (average 18) in each farm. STEC strains were isolated from samples collected from 73 farms, and 138 Stx2k-STEC strains described in this study were detected from 32 farms. The average number of isolates from one farm was ~ 4 in three years (2019-2021).

Samples obtained in 2017 and 2018 (n=732) were collected by local center for disease control and prevention (CDC), the corresponding farms for these samples were missing unfortunately. We have added the farm ID for 138 isolates from 2019 to 2021 in Table S1 as suggested.

Line 294. I suppose the original reference for stx-PCR is Bai et al. (2013, PLOS ONE, doi.org/10.1371/journal.pone.0065537). However, it is not mentioned which primers were used. The primer selection is important for this study and thus they should be mentioned clearly.

Thanks for the reviewer's suggestion. We added this as reference 42.

Line 307. Probably 250 bp would be correct.

Thanks for the reviewer's careful checking. The whole genome was paired-end (2×150 bp) sequenced using Illumina NovaSeq, 350 bp was the fragmented size of DNA sample, we apologize for this mistake and corrected this.

Line 311. What was the criteria for PacBio sequencing? It would be helpful to show PacBio-sequenced isolates in Fig. 3.

The selection for PacBio sequencing is based on sampling sites and molecular traits of strains, as well as the budget for sequencing each year. Due to the high cost of PacBio sequencing, a small number of strains representative of different farms and molecular traits (e.g., serotype, MLST type) were subjected to PacBio sequencing to get closed genomes. We have indicated those PacBio-sequenced isolates in Figure 3 as suggested.

Line 353. Were insertion sites detected by draft genomes?

Insertion sites were detected in 12 complete genomes and 42 draft genomes in this study (Table S1). Among draft genomes, insertion sites were defined only when they were located at the same contig with *stx2k*.

Line 334. Were novel O-types that DebRoy et al. (PLoS One 2016 Vol. 11 Issue 1 Pages e0147434) proposed included?

Yes, our in-house serotyping database included the novel *E. coli* OX serogroups as reviewer mentioned. Besides, our database includes a set of sequences that were recently defined O-types. Given that there were a large number of novel O-types, those undetected in this study were not mentioned in the method to avoid any confusion.

Is it possible to measure the amount of Stx2k by commercial kits?

We have detected the Stx2k production using commercial Duopath® STEC Rapid Test purchased from Germany in our publication where Stx2 was proposed (Yang, Bai et al.2020). Stx2k from most strains were measurable using Vero cell cytotoxicity assay (VCA) and commercial lateral flow immunoassay like Duopath® STEC Rapid Test, though the toxin production level varies significantly among different strains. Due to COVID-19 pandemic, it is presently difficult to purchase commercial kits abroad. We are planning functional study on Stx2k and will measure the amount of Stx2k by commercial kits and other *in vitro* assay in our future studies. We added a statement in discussion (line 303-306).

Again, we sincerely thank all the reviewers for their constructive suggestions. We hope that these modifications will render our paper acceptable for publication in Microbiology Spectrum.

Yours sincerely,

Dr. Yanwen Xiong

State Key Laboratory of Infectious Disease Prevention and Control,

National Institute for Communicable Disease Control and Prevention, China CDC.

July 18, 2022

Prof. Yanwen Xiong
State Key Laboratory for Infectious Disease Prevention and Control, National Institute for Communicable Disease Control and Prevention, China CDC
P.O. Box 5, Changping
Beijing 102206
China

Re: Spectrum01571-22R1 (High prevalence and persistence of Escherichia coli strains producing Shiga toxin 2k subtype in goat herds)

Dear Prof. Yanwen Xiong:

Your manuscript has been accepted, and I am forwarding it to the ASM Journals Department for publication. You will be notified when your proofs are ready to be viewed.

Sincerely,

Sadjia Bekal
Editor, Microbiology Spectrum

Journals Department
Supplemental file 1: Accept
Supplemental file 2: Accept